# Hesitancy toward the Full COVID-19 Vaccination among Kidney, Liver and Lung Transplant Recipients in Italy

**DOI:** 10.3390/vaccines10111899

**Published:** 2022-11-10

**Authors:** Andrea Costantino, Letizia Morlacchi, Maria Francesca Donato, Andrea Gramegna, Elisa Farina, Clara Dibenedetto, Mariarosaria Campise, Matteo Redaelli, Marta Perego, Carlo Alfieri, Francesco Blasi, Pietro Lampertico, Evaldo Favi

**Affiliations:** 1Division of Gastroenterology and Endoscopy, Foundation IRCCS Ca’ Granda Ospedale Maggiore Policlinico, 20214 Milan, Italy; 2Department of Pathophysiology and Transplantation, Università Degli Studi di Milano, 20124 Milan, Italy; 3Respiratory Unit and Adult Cystic Fibrosis Center, Foundation IRCCS Ca’ Granda Ospedale Maggiore Policlinico, 20124 Milan, Italy; 4Division of Gastroenterology and Hepatology, Foundation IRCCS Ca’ Granda Ospedale Maggiore Policlinico, 20122 Milan, Italy; 5Nephrology Dialysis and Transplantation, Foundation IRCCS Ca’ Granda Ospedale Maggiore Policlinico, 20124 Milan, Italy; 6Kidney Transplantation, Foundation IRCCS Ca’ Granda Ospedale Maggiore Policlinico, 20124 Milan, Italy; 7Department of Clinical Sciences and Community Health, Università Degli Studi di Milano, 20124 Milan, Italy; 8CRC “A. M. and A. Migliavacca” Center for Liver Disease, Department of Pathophysiology and Transplantation, Università Degli Studi di Milano, 20124 Milan, Italy

**Keywords:** COVID-19 vaccine, COVID-19, vaccine hesitancy, kidney transplantation, liver transplantation, lung transplantation, solid organ transplant recipients

## Abstract

Background: Coronavirus disease 2019 (COVID-19) vaccination hesitancy is a threat as COVID-19 vaccines have reduced both viral transmission and virus-associated mortality rates, particularly in high-risk subgroups. Solid organ transplant recipients (SOTRs) are particularly vulnerable, as the underlying causes of their organ failure and the chronic immunosuppression are associated with a lower immune response to COVID-19 vaccines, and with an excessive risk of death due to SARS-CoV-2 infection. We aimed to evaluate COVID-19 vaccination hesitancy and its reasons in a population of SOTRs. Methods: All the SOTRs attending our post-transplant clinics were asked to fill in a vaccination status form with specific validated questions related to their willingness to receive a third vaccine dose. In the case of negative answers, the patients were encouraged to explain the reasons for their refusal. Among the SOTRs (1899), 1019 were investigated (53.7%). Results: Overall, 5.01% (51/1019) of the SOTRs raised concerns regarding the future third dose vaccination. In more detail, hesitancy rates were 3.3% (15/453), 4.2% (7/166), and 7.3% (29/400) among the investigated liver, lung, and kidney transplant recipients, respectively (*p* = 0.0018). The main reasons for hesitancy were fear of adverse events (30/51, 58.8%) and perceived lack of efficacy (21/51, 41.2%). Conclusions: Full adherence to ongoing or future vaccination campaigns is crucial to prevent, or at least reduce, COVID-19-related morbidity and mortality in fragile patients. The identification of the reasons influencing COVID-19 vaccination hesitancy in these patients is very important to establish appropriate and targeted patient–doctor communication strategies, and to further implement specific vaccination campaigns.

## 1. Introduction

Since the emergence of the coronavirus disease 2019 (COVID-19), severe acute respiratory syndrome coronavirus 2 (SARS-CoV-2) has caused more than 6.5 million deaths worldwide. However, the prompt application of mass vaccination programs has intensely reduced both viral transmission and virus-associated mortality rates, particularly in high-risk subgroups, such as the elderly, frail, and immunocompromised [1,2,3]. Among the latter group, solid organ transplant recipients (SOTRs) are particularly vulnerable, as the underlying cause of their organ failure and the chronic burden of immunosuppression are associated with a lower immune response to COVID-19 vaccines [4], and with an excessive risk of death due to SARS-CoV-2 infection [5].

The COVID-19 pandemic may be considered as one of the greatest challenges of our time, and its consequences will also affect psychological well-being and may have a harmful impact on mental health. A recent study investigating the main concerns related to COVID-19 among an Italian population identified that the fear about the possible vaccine consequences was more frequent in women, among young adults, and the most frequent among older adults, while the fear of disease and its consequences (as isolation) was more frequent in young adults [6].

Following the release on to the market of effective mRNA COVID-19 vaccines in December 2020 in Italy, the Italian National Health Service has consistently offered free vaccination to all SOTRs. When COVID-19 vaccines were available for SOTRs in Italy (March 2021), the entire cohort of SOTRs (kidney, liver and lung) followed up at Ospedale Maggiore Policlinico, Milan, Italy, was contacted by phone and called for vaccination, with the exception of those who had already voluntarily vaccinated themselves; the same approach was used when the third dose was recommended. In Italy, immunosuppressed patients were offered only mRNA vaccines (either BNT162b2 or mRNA-1273 for all the doses). 

According to the World Health Organization (WHO) Strategic Advisory Group of Experts on Immunization (SAGE), the term ‘vaccination hesitancy’ refers to the ‘delay in acceptance or refusal of vaccination despite availability’ [7]. To date, a limited number of studies have evaluated COVID-19 vaccination hesitancy after SOT. In a previous work, our group has shown that 15% and 6% of liver transplant recipients, respectively, were hesitant or refused the COVID-19 vaccine [8], mainly because of concerns about possible vaccine-induced adverse events.

However, hesitancy may vary depending on the specific transplant population included. 

To the best of our knowledge, only two studies have investigated the vaccine hesitancy among SOTRs (mainly kidney), with very different hesitancy rates (18.6% in the US and 77.2% in China); however, none of these studies has investigated the full adherence to every COVID-19 vaccine dose [9,10]. Since the full adherence to ongoing or future campaigns is crucial to prevent, or at least reduce, COVID-19-related morbidity and mortality, the reasons influencing the hesitancy should be investigated in order to promote an appropriate patient–doctor communication to encourage widespread participation in vaccinations. 

Therefore, the aim of the present single-center cross-sectional study was to assess hesitancy rates toward the third dose of mRNA COVID-19 vaccines in three groups of SOTRs (745 liver, 166 lung, and 988 kidney) followed up at Ospedale Maggiore Policlinico, Milan, Italy. The reasons behind their choices were also evaluated. 

## 2. Materials and Methods

### 2.1. Study Design

In January and February 2022, all SOTRs regularly attending our outpatient post-transplant clinics were asked to fill in a vaccination status form, which included four specific validated questions related to their willingness to receive a third vaccine dose in the near future. A negative answer to the question: “Would you accept a third dose of COVID-19 vaccine tomorrow?” was considered hesitancy against COVID-19 vaccination. In the case of negative answers, the patients were encouraged to explain their refusal (multiple options available); in particular, multiple options for refusal included: a willingness to postpone the third dose in the future, belief that previous doses of vaccination confer sufficient protection, fear of adverse events, lack of confidence in the effectiveness of the vaccination, and willingness to wait for another type of vaccination (other than mRNA-based). 

Since survey research, as with other research, has the potential for a variety of sources of error, the existent strategies to reduce the potential for error were used [11,12]. Of note, before filling in the questionnaire, the patients were required to provide their informed consent. Completion of this survey did not result in any benefit or financial compensation for the patients. COVID-19 vaccines were offered independently of the questionnaire responses. 

### 2.2. Statistical Analysis

Descriptive analysis was performed with the calculation of median and interquartile ranges (IQR) for continuous variables and proportion for categorical variables. Bivariate analyses were conducted using the Student’s *t*-test or Mann–Whitney’s U-test for continuous variables, and chi-square or Fisher’s exact tests for categorical variables. Statistical significance was defined as a 2-tailed α < 0.05. All the statistical analyses were performed using IBM SPSS 22 (SPSS, Inc., Chicago, IL, USA).

## 3. Results

Among the SOTRs (1899), 1019 were investigated (53.7%). Overall, 5.01% (51/1019) of the SOTRs enrolled in the study raised concerns regarding the future third dose vaccination. In more detail, the hesitancy rates were 3.3% (15/453), 4.2% (7/166), and 7.3% (29/400) among the investigated liver, lung, and kidney transplant recipients, respectively (*p* = 0.0018). The main reasons for hesitancy (more than one could be selected) were fear of adverse events (30/51, 58.8%) and perceived lack of efficacy (21/51, 41.2%).

## 4. Discussion

Our data indicate that, although SOTRs are relatively familiar with vaccination as part of the enlistment process or regular post-transplant follow-up (e.g., hepatitis B and flu), 5% may refuse a full COVID-19 vaccination, with a higher rate among kidney recipients compared to the others (lung and liver). The COVID-19 vaccine hesitancy rate among SOTRs is much lower when compared, even if not directly, to the hesitancy to COVID-19 vaccines in the general population worldwide (up to ~70% in some Countries) [13] and when compared to the rate of COVID-19 vaccine hesitancy (31.1%) of the general population in a cohort of the same geographical area (Northern Italy) [14]. Nevertheless, SOTRs’ COVID-19 vaccine hesitancy rate was not so different when indirectly compared to the hesitancy of an Italian cohort of fragile patients (elderly people aged ≥ 65 years old) (5.01% vs. 7.3%) [15].

In contrast, in a previous study, kidney and pancreas transplant recipients under care at a transplant center in the US showed a vaccine hesitancy rate of 18.6%, which was similar to rates reported among the general adult population surveyed across the United States (18.6%). The reasons for hesitancy among these SOTRs were regarding unknown safety of the vaccines in general, a belief that there was a lack of data about the vaccines in transplant recipients, and a lack of trust in the scientific process underlying vaccine development [9].

An anonymous web-based questionnaire was conducted in adult Chinese renal transplant recipients in May 2021. Overall, 813 respondents from 30 provinces all over China participated in the survey, with a response rate of 40.0%. Among the respondents, none of them had a history of SARS-CoV-2 infection and only 5.7% had received any COVID-19 vaccine, while 94.3% had not; 22.8% SOTRs reported that they were willing to get vaccinated, while 65.6% declared that they still hesitate and 11.6% of respondents declared vaccine refusal. Overall, 77.2% SOT participants were categorized as hesitant to the COVID-19 vaccination. Respondents who were unwilling to be vaccinated reported these as the most common reasons: concerns about preexisting comorbidities (77.5%), fear of side effects (59.6%), and the negative advice given by the physician (37.1%) [10].

In our study, the COVID-19 full vaccination hesitancy is much lower compared to these previous reports. If we compare COVID-19 vaccine hesitancy not directly with a population of patients of the same geographical area affected by celiac disease, a chronic immune-mediated disease that is not life threatening in the vast majority of cases, the SOTRs showed a similar refusal rate (4.8%) [16].

Vaccine hesitancy is a complex, global phenomenon and it represents one of the most important criticisms in public health today. Even though it would appear to be a contemporary discussion, the public debate on vaccination is a deeply rooted phenomenon. Many factors have been associated with vaccine hesitancy, including previous negative experiences, education level, healthcare trust, political views, and perception about the importance of vaccination [7]. Regarding COVID-19 vaccines, additional factors may play an important role in vaccine hesitancy: firstly, the speed at which the different vaccines were developed and approved within less than one year, while vaccine development commonly takes years to undergo preclinical stages and clinical trials; secondly, the durability of the immune response following the vaccination and its efficacy to limit the asymptomatic spread [17]; and thirdly, concern and uncertainty during the COVID-19 pandemic raised the spread of misinformation, which extended to affect COVID-19 vaccination [18].

Vaccine hesitancy is, indeed, often determined by incorrect beliefs about health, diseases, and vaccines, which may have been influenced by misinformation. The mass media’s emphasis on the hypothetical side effects of vaccines has triggered waves of misinformation on the safety of vaccines, mainly concerning long-term side effects, the toxicity of adjuvants and preservatives, and the weakening of the immune system [19]. Attitudes toward vaccines can be seen as a continuum, ranging from total acceptance to complete refusal, which is complex and context-specific, varying in different countries. Among the vaccine-hesitant patients, those refusing vaccination are the most difficult to be convinced. There are many factors, such as complacency, convenience, and a lack of confidence in vaccines, that may all contribute to vaccination delay or refusal of one, some, or most vaccines (or to further doses of vaccines) [20]. 

Therefore, it is very important to investigate the hesitancy toward the full COVID-19 vaccination, since some patients who received the initial two doses of mRNA COVID-19 vaccines may, however, question the efficacy and safety of further doses. This could be due to a lower fear of the disease due to a lower perceived severity of the disease compared to the first months of the COVID-19 pandemic.

Patient education may represent the best way to improve vaccinations among SOTRs. In fact, the discrepancy between the real danger and the perceived risk of the COVID-19 vaccine can lead to inappropriate behavior for both at-risk cohorts, such as SOTRs, and also for their relatives. Even when a sufficient level of knowledge is present, messages issued by the physician and effective warnings seem to be necessary.

## 5. Conclusions

Since full adherence to ongoing or future vaccination strategies is crucial to prevent, or at least reduce, COVID-19-related morbidity and mortality, physicians involved in the care of SOTRs should investigate the reasons influencing their hesitancy and promote appropriate patient–doctor communication to encourage widespread participation in vaccination campaigns.

## Data Availability

The datasets generated during and/or analyzed during the current study are not publicly available; however, they are available from the corresponding author upon reasonable request.

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
