# Peer review of "Hesitancy toward the Full COVID-19 Vaccination among Kidney, Liver and Lung Transplant Recipients in Italy"

_vaccines, 2022, doi:10.3390/vaccines10111899_

Round 1

Reviewer 1 Report

This is an interesting, well-written and clearly presented paper. I'm not qualified to assess the statistical analysis. 

Three questions for the authors.

1. In the discussion section, the sentence reads "The COVID-19 vaccine hesitancy rate among SOTRs is much lower when not directly compared to ... the general population." Should "not" be deleted? In other words SOTR patients are less hesitant than the general population, is that correct?

2. Are SOTR patients required to get the initial series for COVID-19 vaccination, or is that voluntary? It would be helpful to clarify this point.

3. If SOTR patients received the initial 2-dose mRNA vaccine, why do some question the safety and efficacy of a third dose? Is there any information or observation you can offer to explain this?

Author Response

We want to thank the reviewer 1 for their compliments and very precise comments to our manuscript.
These are the changes of our manuscript in the revised version:

1) In order to be clearer, we modified the sentence in the discussion according to your precise comment, as it follows: "The COVID-19 vaccine hesitancy rate among SOTRs is much lower when compared, even if not directly, to the hesitancy to COVID-19 vaccines in the general population worldwide (up to ~70% in some Countries)" 

2) We added a sentence in the introduction to precise that patients could get vaccinated voluntarily. but they were invited by us to get vaccinated if they were not yet

3) We added  a small paragraph in the discussion to explain the possible hesitancy to further doses. 

Reviewer 2 Report

Costantino et al. present a prospective investigation of the attitudes of solid organ transplant recipients (SOTR) towards a booster injection of COVID 19 vaccine at a center in Milan, Italy. Over 50% of this center’s SOTR responded (over 1000 patients), and 5% of the patients had concerns about the vaccine. The primary concerns were the perceived lack of efficacy and side effects of the booster immunization. The authors fairly conclude that their findings focus on the issues that should be part of the critical conversation between SOTRs and their physicians concerning future COVID 19 vaccinations.

The manuscript is an easy read, with the goals justified and the results clearly presented. There is only one important issue. Tragically, the authors only obtained oral consent, not written and informed consent. Even worse, the authors did not seek IRB approval prior to their study. Thus, the work does not provide adequate human protections concerning the collection and reporting of this information.

Author Response

We want to thank the reviewer 2 for the compliments and the very precise comment to our manuscript.

Actually, the investigation of patients’ vaccination status (in particular for COVID-19 vaccines in liver and lung SOTRs), their hesitancy towards vaccines and the immunological response to the vaccinations was approved previously by ethics committee (for liver and lung SOTR with the number 422, of Registro delle sperimentazioni 2020/2021, Istituto Nazionale per le Malattie Infettive Lazzaro Spallanzani I.R.C.C.S.) while for vaccination status among kidney SOTR the protocol code 4759-1837/19 of our local ethic commitee).

According to your very precise comment we added the approvation numbers in the Institutional Review Board Statement paragraph at the end of the manuscript. 

Reviewer 3 Report

In this letter, the authors (Dr. Costantino & colleagues) succinctly evaluated the COVID-19 vaccination hesitancy and its reasons in a group of 1019 solid organ transplant recipients. The most important findings indicate that only about 5% (51/1019) of organ transplant recipients raised concerns regarding the future third dose vaccination, while the reasons for hesitancy were fear of of adverse events (58.8%) and perceived lack of efficacy (41.2%). The work is interesting and well written, while the study design is well performed from an experimental design point of view. It will help in understand the reasons influencing the COVID-19 vaccination hesitancy in this class of patients, thus improving the currently employed patient–doctor communication strategies and vaccination campaigns. Please see below several, minor, observations for improving the work that can be accepted unless all these minor observations are well addressed.

1 – acronyms should be mentioned with their complete name the first time being mentioned in the text, e.g., line 25, COVID-19 or line 75 IRCCS
2 – the numbered list in the abstract can be avoided.
3- abstract, the number of patients should be indicated in the methods
4-Line 50 the supporting references should be included PMID:35744711, PMID: 32150360
5-Lines 64-70 these sentences should be moved in to the first section of the introduction
6- I suggest including supporting papers for the statistical methods. For instance, PMID: 26649250 or PMID: 24228679
7-if possible, more information of the surveyed participants should be included. Age, ethnicity, diseases etc…
8-conclusions can be incorporated into the discussion

Author Response

We want to thank the reviewer 3 for the compliments and precise comments to our manuscript.
These are the changes in the revised version:

1) acronym COVID-19 is mentioned with the complete name, we elimintaed IRCCS (which is the acronym for Italian research and care hospitals)

2) we eliminated the numbers in the abstract

3) we indicated the number of patients in the methods

4) we added the supporting references for the paragraph in the introduction

5) actually, we preferred to leave this paragraph in this section of the introduction for a better distinction of the introduction about COVID-19 and its possible consequences on SOTRs and then the introdution on hesitancy. We focused on transplantologists as possible main readers. But thank You for your suggestion. 

6) We added the supporting references in the methods

7) Actually, due to the anonymous nature of the survey we could not add more clinical and socio-demographic information about respondents. 

8) Due to the MDPI indications for manuscripts and indication of the Editor, we should leave a separate paragraph for the conclusions. 

Round 2

Reviewer 2 Report

This will be acceptable, but next time obtain written consent and provide the IRB information in the first submission. 

Author Response

Dear reviewer, 

Thank once again for the very precise correction. 

Best Regards, 

Andrea Costantino
